# The Impact of the COVID-19 Pandemic on the Organisation of Remote Work in IT Companies

Michał Błaszczyk [1,*] , Milan Popović [1,*] , Karolina Zajdel [2] and Radosław Zajdel [1]

1   Department of Computer Science in Economics, Faculty of Economics and Sociology, University of Lodz, 90-255 Lodz, Poland
2   Department of Medical Informatics and Statistics, Medical University of Lodz, 90-419 Lodz, Poland
*   Correspondence: michal.blaszczyk@uni.lodz.pl (M.B.); milan.popovic@uni.lodz.pl (M.P.)

**Abstract:** Some events in world history have affected global social and economic processes significantly. One such event was undoubtedly the commencement of the COVID-19 pandemic. Being in lockdown with access to the Internet and tools that enable remote working enabled and, in fact, forced a change in the form of work to be fully remote, which was previously difficult to imagine in many organisations. As part of this study of the above phenomenon, research was conducted on employees of the IT sector in Poland. An analysis of survey data showed the impact of individual work modes on productivity ratings and collaboration with other team members; additionally, the findings may indicate behaviour changes among employees caused by employer enforcement of changes in work mode, and the importance of the work mode for employees when taking up employment. Although the end of the COVID-19 pandemic has been officially announced in Poland, its impact on the labour market has been significant. The present study shows the popularisation of remote working and a change in attitude towards this form of performing professional duties.

**Keywords:** remote work; IT; employees; work organisation; COVID-19

## 1. Introduction

For two years, the world as we knew it changed beyond recognition. A global coronavirus pandemic caused the deaths of hundreds of thousands of people [1,2] and a global physical and mental health crisis [3–5]. It has affected not only the area of healthcare, for example, by popularising telemedicine [6–8], but also multiple other areas, including the labour market [9].

Coronavirus disease 2019 (COVID-19), the sickness caused by severe acute respiratory syndrome coronavirus 2 [10] (SARS-CoV-2), was first identified in Wuhan in December 2019 [11]. The virus quickly expanded in China [12] and then quickly advanced to become a worldwide pandemic [12]. The pandemic in Poland officially began on 20 March 2020, with the Regulation of the Minister of Health announcing the epidemic in the territory of the Republic of Poland [13]. With this regulation, the first restrictions related to the prevention of coronavirus transmission came into effect [14,15]; among them was a prohibition on relocation except in justified situations. Educational institutions, shopping malls, hotels, and stores were closed [15]. Such restrictions were intended to ensure that social distancing could be maintained. After the holiday break, the country was divided into zones, with restrictions based on the number of people infected with COVID-19 in a given county [16].

The pandemic situation in Poland became worse in March 2021 [17,18]. There was a significant increase in infections, followed by increases in bed occupancy in hospitals and ventilator occupancy, as well as an increase in deaths. New safety rules were introduced. Large furniture and construction stores covering more than 2000 square meters were closed. Malls and shopping centres were also closed, except for grocery stores, pharmacies, and bookstores [18]. Hairdressing and beauty salons were also unable to operate. The obligation

to cover one's mouth and nose and keep a distance remained unchanged. Nurseries and kindergartens were available for the children of medical personnel and uniformed service employees only. Moreover, organised amateur sports were forbidden, and only those preparing for professional sports were able to train [17,18]. Due to exacerbations related to the epidemiological state, thousands of workers were immediately required to work from their homes [19], forcing a change in the remote work model.

Remote work itself is a concept that emerged before the development of the COVID-19 pandemic. In 1973, the first experimental telework project was conducted [20,21]. Remote work is defined as work that allows its results to be transmitted by methods, such as telecommunications and computers, instead of by the physical movement of people [20]. The European Union framework agreement on telework defines telework as "a form of organising and/or performing work, using information technology, in the context of an employment contract/relationship, where work, which could also be performed at the employer's premises, is carried out away from those premises on a regular basis" [22]. In Polish law, teleworking appeared with the amendment to the Labour Code of August 2007 [23], with the justification that teleworking enables employees to combine professional and private activities. As a form of flexible employment, it has a positive impact on the professional activities of people who live in a place away from the workplace, are disabled, perform family duties, combine work and study, or are employed in more than one workplace [23]. Teleworking was defined as work performed regularly outside the workplace, with the use of an electronic means of communication; therefore, a teleworker is an employee who performs work in the abovementioned way and communicates their results with their employer, particularly via an electronic means of communication [24].

Remote work before the COVID-19 pandemic was not yet defined in the Labour Code. However, the popularisation of this form of work in connection with the COVID-19 pandemic and the related quarantine rules contributed to the need to legally regulate it [25]. Therefore, among other things, the Coronavirus Act defined and made it possible to organise remote work. The legislature provided rules for organising such work, and addressing issues related to ensuring employees' health and safety and those related to ensuring the safety of the data which may be necessary to perform such remote work [26]. Some researchers anticipated remote work as a synonym for telework [27]. However, remote work as introduced in the Coronavirus Act does not constitute telework within the meaning of the regulations, as remote work is a broader concept than teleworking [24]. The new regulations state that, among other conditions, in order to prevent COVID-19, an employer may instruct an employee to perform, for a specified period of time, the work specified in the employment contract outside the place of its regular performance, i.e., remote work [28]. The law does not specify either the maximum duration of such work or what is meant by "work outside the place of its regular work". The new regulations also do not specify what such remote work would consist of, nor do they indicate whether an order for remote work can be in response to an employee's request.

Remote work does not mean that the employee is out of the employer's control. The employer can give the employee ongoing instructions and recommend specific tasks during working hours. They can also control whether the employee actually performs work during this time. To this end, the employee may be required to be constantly "on the phone" or "on email" during working hours [26].

## 2. Literature Review

A number of studies on remote work have been published, addressing the legal [29,30], economic [31,32], and social aspects [33,34] of the changes brought about by the COVID-19 pandemic. The year 2020 has undoubtedly intensified the need to transform the traditional "in-office" workplace to "work from home" or "hybrid working" [35]. A review of both past and present theoretical background and empirical research surrounding changes in working mode and perceptions of productivity is crucial for further development in this field. Remote working has enforced a consolidation of professional and personal activities;

therefore, it can be one of benefit for employees, as well as generating savings for employers. However, the COVID-19 pandemic also resulted in a significant number of factors that made it difficult to perform daily tasks for work. Such challenges included isolation and a lack of social contact. Increased levels of stress, combined with continued remote working, can cause disruption to employees' wellbeing [36]. Many of the factors identified in the literature as negative consequences of remote work have gained in intensity during the pandemic. These include the lack of contact with co-workers [37], blurring of the work–life boundary [38], the feeling of being constantly at work [37–40], family–work conflict, and social isolation [41]. It is considered that, in order to separate work and personal life and to protect worker health and safety from the dangers of an overly connected work culture, specific measures, including the right to disconnect, must be taken in the near future [42]. While remote work is associated with these disadvantages, it also brings certain benefits, such as money and time saving, removal of the distance barrier, no office distractions [43], and flexibility [44]. It was also noticed that remote work perceptions depend on gender; when considering the use of ICT and digital communication tools in the work process, women may not stay connected after formal working hours to the same extent as their male counterparts. Disconnection could be more crucial for women in terms of maintaining work–life balance [45,46]. Moreover, it has been pointed out that involvement in remote working while living with young children plays a key role in the relationship between overall work productivity and remote working productivity. The moderating effect of living with young children highlighted that the effect of remote work involvement on remote work productivity was found to be more favourable for workers not living with young children than for those cohabiting with children [47]. Researchers also found that fear of COVID-19 is positively associated with higher levels of productivity and engagement. Those who were emotionally affected by the COVID-19 pandemic also reported being more productive and motivated when working from home. This may lead one to conclude that this way of working may play a protective, anxiety-relieving role for workers, because they were not asked to physically go to work and thus avoided exposure to possible contagion by staying at home [41,48]. On the other hand, concerns about COVID-19 infection are decreasing; this was highlighted by a comparison between 2020 and 2022 data [49]. Consequently, the impact of the COVID-19-related fear on perceptions of remote work may be reduced compared with that at the start of the pandemic. It was also observed that perceptions of lower productivity during remote work were associated with the increasing age of workers. It may be related to the difficulties older workers may face in using technological tools and their potentially lower ability to adapt to rapid changes [41].

Businesses are set to accelerate the digitalization of work processes, and expansion of remote work, and the automation of tasks within their organizations. It is urgent to address the disruption underway both by supporting and retraining displaced workers and by monitoring the emergence of new opportunities in the labour market [50]. Nevertheless, based on the data from the World Economic Forum in 2020, nearly 80% of business leaders expected a negative impact of remote and hybrid work on employee productivity; nearly one in four expected a strong negative impact; and only one in six thought it would have no impact or a positive impact on productivity [51]. Furthermore, a study on team virtuality, conducted on a sample of workers from different countries, found that social isolation, in both its facets of physical and informational isolation, negatively predicted job satisfaction [52], which was found to be negatively related to the perception of remote work productivity [53]. On the other hand, some research shows how respondents perceived their own productivity to be better during COVID-19, despite the significant disruptions to their non-work life [54].

## 3. Materials and Methods

The changes in the labour market caused by the COVID-19 pandemic are unquestionable and clearly noticeable; therefore, the authors decided to examine not only the fact of COVID-19's impact on the labour market, but also its scale in a specific area of IT companies.

In view of this, the authors found that studies in this area are inconsistent [29–41,51–54] and decided to determine how the remote work is perceived by IT-sector employees in Poland. Consequently, the following hypothesis was formulated.

**Hypothesis 1 (H1).** *Most employees in the IT sector in Poland perceive the change in working mode to remote as positive and their perceived productivity increases.*

This study focuses on employment in the IT market in the broadest sense; therefore, it is reasonable to review the demand for IT professionals before and after the pandemic. The year 2019 marked the end of a decade of very strong labour market growth and employer interest in new hires [55]. The annual number of advertisements on Pracuj.pl, the most popular job-search portal in Poland [56], more than doubled between 2010 and 2019 with an increase of 132%. Demand in recent years, prior to the COVID-19 pandemic, remained stable for IT experts, with a 15% share [55]. Following the announcement of the end of the pandemic in Poland, IT specialists were the most in demand, for the first time in the history of the survey. They were targeted by nearly one in four offers on Pracuj.pl (24%) in the first half of 2022, a 60% increase compared with their offers before the COVID-19 pandemic [57].

A questionnaire was built to collect data online from IT sector employees (see Supplementary Materials). An online format was selected due to the nationwide character of the study. The study comprised 187 participants. The respondents voluntarily accessed an online MS Forms questionnaire through a link published via email to selected companies from the IT sector. The research was promoted as an anonymous and confidential. Due to the online nature of the form, data were collected by snowball sampling [58], so authors did not have full control over the pool of respondents. The nature of snowball sampling is such that it cannot be considered as a representative sample [59]; the authors accounted for this characteristic in the analysis.

The questionnaire consisted of two parts. In the first, respondents were asked about their model of work, a self-evaluation of productivity, and whether they had changed employer. In the second part, respondents were asked several questions concerning their demographic data and their employers' data. In order to verify the representativeness of the sample in relation to the IT labour market in Poland, the data obtained from respondents were compared with a survey carried out in the same year by the BullDogJob.pl job portal [60], in which the sample size was 6482 respondents; here, only respondents who were positively verified for data quality were analysed. Recruitment for the BullDogJob.pl survey took place via the website, social media channels, online publishers, and email. The survey was fully anonymous [60]. Two methods were used: similarity indices for the structures of the analysed communities and the chi-square test of concordance. For all factors, $\alpha = 0.05$ was taken as the significance level. The similarity index of the structures was calculated based on the following formula [61]:

$$\omega_p = \sum_{i=1}^{k} min(\omega_{1i}, \omega_{2i}) \tag{1}$$

where $0 \leq \omega_p \leq 1$ and $i = 1, \ldots, k$; $k$—number of classes; $\omega_{1i}$, $\omega_{2i}$—indicators of the structures of the analysed communities for the class number, $i$. The closer the resulting index is to 1, the more similar the structures of the studied communities are [61]. A chi-square test was then carried out to verify representativeness further on. With this test, the authors decided to verify the hypothesis concerning the distribution of the analysed populations. For the purposes of the test, it was verified whether respondents who completed the survey were characterized by the same distribution (with respect to individual characteristics) as

the IT community survey respondents. In the chi-square test of concordance, the statistics were calculated with the following formula [62]:

$$\chi^2 = \sum_{i=1}^{k} \frac{(n_i - \hat{n}_i)^2}{\hat{n}_i} \qquad (2)$$

where $i = 1, \ldots, k$; $k$—number of classes; $n_i$—the size of the class numbered $i$; $\hat{n}_i$—the hypothetical size of a class numbered $i$. This was calculated with the following formula: $\hat{n}_i = n \cdot p_i$; here, $p_i$ is the hypothetical probability that the attribute under consideration takes values in the class, $i$. In order to verify the claim, it is necessary to read the critical value for the test from the array of the chi-squared distribution [63] with k – r – 1 degrees of freedom, where r is the number of estimated parameters of the feature distribution [64].

## 4. Results

The study comprised two stages. In the first, the representativeness of the collected data was assessed, as described above. In the second, statistical analyses of the obtained results to verify hypothesis were carried out.

Since it was not possible to verify the representativeness of the sample in relation to the population, the authors decided to compare the sample under study with a larger survey conducted in the same year by the BullDogJob.pl portal in Poland [60]. Accordingly, the authors of the study assumed that a larger sample size correlates with greater sample representativeness [65].

First, the authors analysed the distribution of the study sample of structures to the respondents' positions, with only their IT-related position taken into consideration. Table 1 presents the similarity index of structures between the evaluated data and the data of the BullDogJob.pl survey, that had a much higher number of respondents. Based on the analysis, where the number of classes was 5, it appeared that the similarity of structures was high, with a level of 0.84.

**Table 1.** Similarity index of structures to respondent position (only IT-related position taken into consideration).

| Position | $\omega_1$ | $\omega_2$ | $min(\omega_{1i}, \omega_{2i})$ |
|---|---|---|---|
| Programmer | 0.45 | 0.57 | 0.45 |
| Tester | 0.23 | 0.18 | 0.18 |
| Analyst | 0.08 | 0.06 | 0.06 |
| Project/team manager | 0.18 | 0.09 | 0.09 |
| Network administrator/engineer | 0.06 | 0.10 | 0.06 |
| **TOTAL** | **1.00** | **1.00** | **0.84** |

On the other hand, taking into account the fact that the number of classes was 5, no parameter was estimated, and the significance level was assumed as $\alpha = 0.05$; then, with the number of degrees of freedom at 4, the critical value, $\chi^2_{4;0.05}$, was 9.49 [63]. Since the calculated value of the statistic $\chi^2 = 24.34$ exceeded the critical value and thus belonged in the right-hand critical set in the chi-squared distribution, hypothesis H0—that respondents who completed the survey were characterized by the same distribution with respect to individual characteristics as the IT community survey respondents—should be rejected in favour of hypothesis H1; this hypothesis states that respondents who completed the survey were characterized by a different distribution with respect to individual characteristics than respondents to the IT community survey. The results of the chi-square compatibility test to respondent position are presented in Table 2.

**Table 2.** Chi-square compatibility test to respondent position (only IT-related position taken into consideration).

| Position | $p_i$ | $n_i$ | $\hat{n}_i$ | $\dfrac{(n_i - \hat{n}_i)^2}{\hat{n}_i}$ |
|---|---|---|---|---|
| Programmer | 0.57 | 74 | 94.05 | 4.27 |
| Tester | 0.18 | 38 | 29.7 | 2.32 |
| Analyst | 0.06 | 14 | 9.9 | 1.70 |
| Project/team manager | 0.09 | 29 | 14.85 | 13.48 |
| Network administrator/engineer | 0.10 | 10 | 16.5 | 2.56 |
| **TOTAL** | **1.00** | **165** | **165** | **24.34** |

The authors also analysed the distribution of the study sample of structures for the resident towns and cities of the respondents. Table 3 presents the similarity index of structures between the evaluated data and the data representing a much higher number of respondents. Based on the analysis, where the number of classes was 3, it appeared that the similarity of structures was high, with a level of 0.88. On the other hand, taking into account the fact that the number of classes was 3, no parameter was estimated, and the significance level was assumed as $\alpha = 0.05$; then, with the number of degrees of freedom as 2, the critical value, $\chi^2_{2;0.05}$, was 5.99 [63]. Since the calculated value of the statistic $\chi^2 = 21.02$ exceeded the critical value and thus belonged to the right-hand critical set in the chi-squared distribution, hypothesis H0—that respondents who completed the survey were characterized by the same distribution, with respect to individual characteristics, as the IT community survey respondents—should be rejected in favour of hypothesis H1; this hypothesis states that respondents who completed the survey were characterized by a different distribution, with respect to individual characteristics, than respondents to the IT community survey. The results of the similarity index and chi-square compatibility test for the resident towns and cities of the respondents are presented in Tables 3 and 4, respectively.

**Table 3.** Similarity index of structures for the resident towns and cities of respondents.

| Residence | $\omega_1$ | $\omega_2$ | $min(\omega_{1i}, \omega_{2i})$ |
|---|---|---|---|
| Cities over 500 thousand inhabitants | 0.51 | 0.63 | 0.51 |
| Cities 100–500 thousand inhabitants | 0.30 | 0.27 | 0.27 |
| Other | 0.19 | 0.10 | 0.10 |
| **TOTAL** | **1.00** | **1.00** | **0.88** |

**Table 4.** Chi-square compatibility test for the resident towns and cities of respondents.

| Residence | $p_i$ | $n_i$ | $\hat{n}_i$ | $\dfrac{(n_i - \hat{n}_i)^2}{\hat{n}_i}$ |
|---|---|---|---|---|
| Cities over 500 thousand inhabitants | 0.63 | 95 | 117.81 | 4.42 |
| Cities 100–500 thousand inhabitants | 0.27 | 56 | 50.49 | 0.60 |
| Other | 0.10 | 36 | 18.70 | 16.00 |
| **TOTAL** | **1.00** | **187** | **187** | **21.02** |

The authors also analysed the distribution of the study sample for structures of working experience. Table 5 presents the similarity index of structures between the evaluated data and the data representing a much higher number of respondents. Based on the analysis, where the number of classes was 3, it appeared that the similarity of structures was high, with a level of 0.89. The results of a similarity index of the structures of work experience are presented in Table 5.

**Table 5.** Similarity index of structures of work experience.

| Work Experience | $\omega_1$ | $\omega_2$ | $min(\omega_{1i},\omega_{2i})$ |
|---|---|---|---|
| Up to 2 years (junior) | 0.14 | 0.20 | 0.14 |
| 2–5 years (mid/regular) | 0.41 | 0.46 | 0.41 |
| More than 5 years (senior) | 0.45 | 0.34 | 0.34 |
| **TOTAL** | **1.00** | **1.00** | **0.89** |

On the other hand, taking into account the fact that the number of classes was 3, no parameter was estimated, and the significance level was assumed as $\alpha = 0.05$; then, with the number of degrees of freedom of 2, the critical value $\chi^2_{2;0.05}$, was 5.99 [63]. Since the calculated value of the statistic $\chi^2 = 11.86$ exceeded the critical value and thus belonged to the right-hand critical set in the chi-squared distribution, hypothesis H0—that respondents who completed the survey were characterized by the same distribution, with respect to individual characteristics, as the IT community survey respondents—should be rejected in favour of hypothesis H1; this hypothesis states that respondents who completed the survey were characterized by a different distribution, with respect to individual characteristics, than respondents to the IT community survey. The results of the chi-square compatibility test for work experience are presented in Table 6.

**Table 6.** Chi-square compatibility test for work experience.

| Position | $p_i$ | $n_i$ | $\hat{n}_i$ | $\frac{(n_i-\hat{n}_i)^2}{\hat{n}_i}$ |
|---|---|---|---|---|
| Up to 2 years (junior) | 0.20 | 26 | 37.40 | 3.47 |
| 2–5 years (mid/regular) | 0.46 | 76 | 86.02 | 1.17 |
| More than 5 years (senior) | 0.34 | 85 | 63.58 | 7.22 |
| **TOTAL** | **1.00** | **187** | **187** | **11.86** |

The authors also analysed the distribution in the study sample for structures of the number of employees in the surveyed employees' companies. Table 7 presents the similarity index of structures between the evaluated data and the data representing a much higher number of respondents [66]. Based on the analysis, where the number of classes was 4, it appears that the similarity of structures was medium, with a level of 0.79. The results of the similarity index for the structures of the number of employees in the surveyed employees' companies are presented in Table 7.

**Table 7.** Similarity index for structures of the number of employees in the surveyed employees' companies.

| Number of Employees | $\omega_1$ | $\omega_2$ | $min(\omega_{1i},\omega_{2i})$ |
|---|---|---|---|
| 2–9 employees | 0.01 | 0.09 | 0.01 |
| 10–49 employees | 0.04 | 0.16 | 0.04 |
| 50–249 employees | 0.18 | 0.20 | 0.18 |
| 250 or more employees | 0.76 | 0.56 | 0.56 |
| **TOTAL** | **1.00** | **1.00** | **0.79** |

Taking into consideration the fact that the number of classes was 4, no parameter was estimated, and the significance level was assumed to be $\alpha = 0.05$; then, with the number of degrees of freedom as 4, the critical value, $\chi^2_{4;0.05}$, was 9.49 [63]. Since the calculated value of the statistic $\chi^2 = 43.21$ exceeded the critical value and thus belonged to the right-hand critical set in the chi-squared distribution, hypothesis H0—that respondents who completed the survey were characterized by the same distribution, with respect to individual characteristics, as the IT community survey respondents—should be rejected in favour of hypothesis H1; this hypothesis states that respondents who completed the survey were characterized by a different distribution, with respect to individual characteristics,

than respondents to the IT community survey. The results of the chi-square compatibility test for the number of employees in the surveyed employees' companies are presented in Table 8.

**Table 8.** Chi-square compatibility test for the number of employees in the surveyed employees' companies.

| Number of Employees | $p_i$ | $n_i$ | $\hat{n}_i$ | $\frac{(n_i - \hat{n}_i)^2}{\hat{n}_i}$ |
|---|---|---|---|---|
| 2–9 employees | 0.09 | 2 | 16.26 | 12.51 |
| 10–49 employees | 0.16 | 8 | 29.51 | 15.68 |
| 50–249 employees | 0.20 | 34 | 37.34 | 0.30 |
| 250 or more employees | 0.56 | 143 | 103.89 | 14.72 |
| **TOTAL** | **1.00** | **187** | **187** | **43.21** |

The second part of the study is related to respondents' statements about remote working.

Initially, a model of respondents' work before the COVID-19 pandemic was examined. The vast majority of respondents (82%) worked in stationary or hybrid conditions, with a predominance of stationary work, with as many as 58% of the respondents working only stationary. Only 8% of the respondents worked fully remotely.

The question of whether the COVID-19 pandemic changed the work model of respondents was addressed. The results showed that, in 80% of cases, respondents' work models did change.

For the respondents with changes in their work model, we examined the directions of these changes. The results showed that, for a significant majority of respondents (86%), their work model changed to remote or hybrid, with a predominance of remote work; meanwhile, the majority (as many as 56% of respondents) confirmed a change in their work mode to only remote work. A small percentage (only 12% of respondents) were in favour of changing their work mode to hybrid, with a predominant preference for stationary work.

Another issue concerns the perceptions of respondents surrounding their changing work models and relationships with their colleagues. In this area, the opinions of the respondents were highly varied. Positive feelings surrounding changes were reported by nearly one in four respondents (27%), and negative feelings were reported by one in five (20%).

Respondents' feeling about their own productivity in remote and stationary work, as well as in hybrid models, was also examined. Interestingly, the vast majority (70%) reported greater productivity when working remotely compared with stationary work.

Additionally, the respondents were asked which model of work they feel is the best fit for them. Almost one in three respondents (35%) reported that they prefer fully remote work; similarly, one in three respondents declared preferring the hybrid model with a predominance of remote work (35%); the hybrid model with a predominance of stationary work was preferred by only almost one in five respondents (21%).

The respondents were also asked how they might behave in a situation in which their employer wanted to force them to change their mode of work. Almost half of the respondents (48%) replied in the affirmative.

At the end, the respondents were asked whether, in the hypothetical case of a change of employer, they would take into account the model of the offered work. Here, the vast majority (75%) reported that their decision would be influenced by the mode of work offered by the employer. The results of the survey are presented in Table 9.

**Table 9.** Survey results summary.

| | | |
|---|---|---|
| **Work Model before the COVID-19 Pandemic** | | |
| Fully stationary work | 109 | 58% |
| Hybrid work with a predominance of stationary work | 44 | 24% |
| Hybrid work with a predominance of remote work | 11 | 6% |
| Fully remote work | 15 | 8% |
| I did not work before the COVID-19 pandemic | 8 | 4% |
| **Work model change due to the COVID-19 pandemic** | | |
| Changed | 144 | 80% |
| Unchanged | 30 | 17% |
| It's hard to say | 5 | 3% |
| **New work model, if changed** | | |
| Fully stationary operation | 3 | 2% |
| Hybrid work with a predominance of stationary work | 17 | 12% |
| Hybrid work with a predominance of remote work | 44 | 31% |
| Fully remote work | 80 | 56% |
| **Opinion of the change of the work model on cooperation with other employees in the company** | | |
| Positive (now it is better than it was before the COVID-19 pandemic) | 39 | 27% |
| Negative (before the COVID-19 pandemic it was better) | 28 | 19% |
| None (I did not feel the change) | 49 | 34% |
| Hard to say/not applicable | 28 | 19% |
| **Own productivity rating while working remotely compared to stationary work** | | |
| Definitely higher when working remotely | 53 | 30% |
| Moderately higher when working remotely | 73 | 41% |
| Moderately higher during stationary work | 30 | 17% |
| Definitely higher during stationary work | 11 | 6% |
| I don't have a comparison/I don't know | 12 | 7% |
| **Work model that fit the best to respondent in their opinion** | | |
| Fully stationary work | 12 | 6% |
| Hybrid work with a predominance of stationary work | 40 | 21% |
| Hybrid work with a predominance of remote work | 66 | 35% |
| Fully remote work | 65 | 35% |
| I don't have a comparison/I don't know | 4 | 2% |
| **Considering change of employer if work model is forced to be changed** | | |
| Yes | 90 | 48% |
| No | 44 | 24% |
| It's hard to say | 53 | 28% |
| **Employment dependency on the mode of work offered** | | |
| Yes | 141 | 75% |
| No | 22 | 12% |
| It's hard to say | 24 | 13% |

## 5. Discussion

The results obtained showed that the similarity index of structures for all characteristics oscillates around 0.85, while in all cases the chi-square test indicated a lack of representativeness in comparison with the survey conducted by a popular work-offering portal in Poland. As representativeness was not tested against the population due to the impossibility of obtaining the necessary data, and was only compared with a larger survey's sample, the hypothesis that the data obtained in the survey were not representative cannot be validated. However, this hypothesis cannot be rejected either. Accordingly, the results obtained were analysed in aggregate, with no breakdown by individual characteristics.

The aim of the study was not to show that the working model has changed, because that it has been already proven in the literature [67–69]. The aim was to find out whether this change has, firstly, made a difference in terms of productivity and, secondly, to determine employee perceptions of remote working. As it was not possible for the first objective to examine real productivity according to the working model, the perceptions of IT employees concerning their own productivity were examined. The results of the survey clearly indicated that the vast majority, up to 70% of respondents, felt they were more productive when working remotely than when working from home. It should be noted that people who switched to the remote or hybrid work model with a predominance of remote work experienced a positive impact in their cooperation with other employees (27%) or did not feel such a change at all (35%); only 19% of respondents reported a negative change in cooperation.

In addition, the impact of remote working, propagated as a result of the COVID-19 pandemic, has been researched by various research centres around the world, including Harvard University and New York University [70]; such studies have revealed interesting trends regarding electronic activity and communication. The literature and our findings suggest that employers' concerns about the negative impact of remote working on productivity appears to be baseless. On the other hand, the observations on working time described above, especially the increase in the intensity of emailing outside of normal working hours, may confirm the researchers' fears that remote working will be perceived by employees as a negative phenomenon, in that it blurs the boundary between work and private life [37–40].

However, the survey found that almost one in three respondents prefer to work fully remotely. Similarly, a preference for a hybrid model with a predominance of remote working was reported, while a hybrid model with a predominance of stationary working was preferred by almost one in five respondents. It is noteworthy that a minority of respondents, only 7%, prefer to work exclusively at the employer's location. Furthermore, among those who switched to a remote or hybrid mode of work due to the pandemic, when asked whether they would leave if their employer forced a change of work mode, up to 58% reported that they would consider changing their employer.

In summary, the COVID-19 pandemic influenced employees' perceptions of their working model; their perceptions were found to influence employee dependency on further employment with their current employer, with the working model becoming one of the primary factors in choosing a new employer. Such a situation is particularly relevant when unemployment is at a very low level [71]. Considering the abovementioned data, it was concluded that—despite the disadvantages of remote work indicated here—at least among the IT sector employees, the majority considered a remote model of work to be the most preferable.

## 6. Conclusions

Due to the pandemic, a majority of Europeans worked from home (at least partially) [72]. In Poland, the end of the pandemic state has been officially declared [73,74]; however, its impact on the labour market is undoubtedly significant. The present study revealed the popularisation of remote working and a change in general approach to this form of work.

As expected, results showed that the COVID-19 pandemic had a significant impact on changing work patterns in the IT business sector. Before the COVID-19 pandemic, 82% of employees worked in stationary or hybrid conditions, with a predominance of stationary work. As a result of the pandemic, up to 74% of employees changed their working mode to remote or hybrid, with a predominance of remote working. Despite the concerns raised in the literature review [29–54], the findings proved a positive impact on employees' perception of their own productivity when working remotely. Moreover, although many of negative consequences of remote work have been identified in the literature [37–41,48,51–54], the outcomes of this study proved that this working model is

considered as the most suitable for employees. This allowed to verify hypothesis, that a change to a remote working model would positively affect employees' perceptions of productivity ratings, and that after the COVID-19 pandemic, the majority of IT employees would have a positive perception of the change to a remote working model. In both cases, these claims can be considered as appropriate. Additionally, it was found that a significant proportion of the participants in this study reported a willingness to change their employer should they be forced to change their working model. Thus, it may concluded that, in the case of the respondent group, the advantages of remote work outweigh the disadvantages, or that the disadvantages are irrelevant; such findings have been shown in other studies [44,75]. These findings may be challenging for employers. In this context, it would be interesting to examine how the labour market has changed in employers' perspectives as a result of COVID-19: whether they would like to return to a stationary work model once the pandemic is over and how they could convince employees to accept this change.

In summary, the COVID-19 pandemic itself should be considered not only in relation to healthcare, but also in a socioeconomic and psychological [76] context. The changes that have taken place due to COVID-19 will last longer than the restrictions imposed to control the virus; the need for digital transformation will increase in coming years [77], along with changing perceptions and demands surrounding remote working models.

**Supplementary Materials:** The following supporting information can be downloaded at: https://www.mdpi.com/article/10.3390/su142013373/s1. File S1: The study questionnaire.

**Author Contributions:** Conceptualization, M.B., M.P., K.Z. and R.Z.; methodology, M.B. and M.P.; software, M.B. and M.P.; validation, M.B., M.P., K.Z. and R.Z.; formal analysis, M.B. and M.P.; investigation, M.B. and M.P.; resources, M.B. and M.P.; data curation, M.B. and M.P.; writing—original draft preparation, M.B. and M.P.; writing—review and editing, M.B. and M.P.; visualization, M.B. and M.P.; supervision, M.B. and M.P.; project administration, M.B. and M.P.; funding acquisition, R.Z. All authors have read and agreed to the published version of the manuscript.

**Funding:** This research received no external funding.

**Institutional Review Board Statement:** Ethical review and approval were waived for this study, because ethical committee reviews are only required when biological material (biological, medical, chemical, physical) is collected or when scientific research interferes with the human psyche; the present study covers areas that do not qualify for ethics committee's consideration. The study was anonymous and voluntary, without any compensation for the participants. Approval for the study was not required in accordance with the University of Lodz Research Ethics Committee (https://www.uni.lodz.pl/fileadmin/user_upload/rules_of_procedure_of_the_ul_research_ethics_committee.pdf, accessed on 29 July 2022).

**Informed Consent Statement:** Informed consent was obtained from all subjects involved in the study.

**Data Availability Statement:** Data are contained within the article and the Supplementary Materials.

**Conflicts of Interest:** The authors declare no conflict of interest.

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
