# Peer review of "The Impact of the COVID-19 Pandemic on the Organisation of Remote Work in IT Companies"

_sustainability, doi:10.3390/su142013373_

Round 1
Reviewer 1 Report
The paper explores impact of the COVID-19 pandemic on the organization of remote work in IT companies.
1. The introduction section should put the current study in the context of previous studies. It does not do that properly.
2. Also the contribution of the study should be emphisized. Why is this study valuable? I find that the research question is quite simple and has been proven before.
3. The impact of COVID 19 cannot be proved with the methdology chosen. The methods prove that distribution has changed during COVID 19, but the statistical analysis does not prove that. Has COVID 19 impact somehow been conceptualized and operationalized in the study?
4. The paper is also based on a qualitative study. The second part of the study is related to respondents' statements about remote working. Parts of the paper claim that other things have been hypothesized, and than proved in qualitative study. This is not a proper framing of the methods.
This part of paper is a simple survey, and has not much scientific merit.
5. Overall the style and language are fine, and some proofreading is needed, but the major issue are the methods that do not respond to testing the hypotheses.
Author Response
Dear Reviewer,
Thank you very much for your rightful comments on our work, which we have taken into account as follows.
- We put the current study in the context of previous studies in the introduction section, which give us opportunity to better explanation of the paper aim.
- We realize that the impact of the pandemic on the change of the work model is obvious and has been proved before, therefore we have decided to examine not the fact of COVID-19's impact on the labor market, but its scale in a specific area of IT companies and perception of the productivity.
- Our study, through the respondents' answers, showed the scale of COVID-19's impact on the labor market, as described in the statistical studies. As stated earlier, our goal was not the significance of the impact, but its scale.
- We have changed the form of the hypothesis so that it can be proven straightforwardly using qualitative data.
The article has been verified by a proofreader, so grammatical errors have been removed.
Thank you very much for your insightful review, which allowed us to make changes to improve the quality of our article. We are open to any further feedback.
Kind regards,
Milan Popović
Michał Błaszczyk
Karolina Zajdel
Radosław Zajdel

Reviewer 2 Report
The choice of topics for the study is appropriate and interesting, but there are a number of areas for improvement. The number of literature sources used in the study is adequate, but the introduction of the study is short for the article's size. I request that the authors introduce the topic better, providing a better literary foundation for it. The methodological background of the study also needs to be improved along with the results. In the results section, there are a number of figures showing the distribution of responses to a question by respondents. It would be worthwhile to accompany these with a more in-depth statistical analysis than just showing the distribution of responses. I recommend that the authors indicate the sources after each figure and table, together with the methodologies used. It would also be necessary to include a conclusions section at the end of the paper instead of/beside the discussion. I suggest that the hypotheses should be numbered 1 and 2 and that a summary table in the conclusions section should show how they have been proved for better followability. Overall, I suggest expanding the introduction, improving the methodology used, and presenting the results in more depth instead of/alongside descriptive figures.
Author Response
Dear Reviewer,
Thank you very much for your rightful comments on our work, which we have taken into account as follows.
We put the current study in the context of previous studies in the introduction section, which give us oportunity to better explanation of the paper aim. We have extended the introduction about 30%. Although, the hypothesis that the data obtained in the survey are not representative cannot be validated. However, this hypothesis cannot be rejected either. Accordingly, the results obtained were analyzed in aggregate. Consequently, we were not in a position to conduct in-depth statistical analysis by individual characteristics. We did not indicate the sources after each figure and table, since all of figures and tables are elaborated by authors and this is consistent with the template and other articles published in the journal. The conclusion section has been included after the discussion section. We have changed the form of the hypothesis so that it can be proven straightforwardly using qualitative data. The article has been verified by a proofreader so, grammatical errors have been removed.
Thank you very much for your insightful review, which allowed us to make changes to improve the quality of our article. We are open to any further feedback.
Kind regards,
Milan Popović
Michał Błaszczyk
Karolina Zajdel
Radosław Zajdel

Reviewer 3 Report
The authors have conducted an interesting research titled “The impact of the COVID-19 pandemic on the organization of remote work in IT companies”. In general, I found their research reliable and insightful to contribute to COVID-related research. I can recommend it for publication after a revision on some parts. In this regard, please kindly find my comments as follows.
1- The introduction section is not well defined and does not read well. The main research gap, research focus, research objective, and contribution of the research are not clear at all in the introduction section. I highly recommend the authors thoroughly rewrite this section and try to come up with a clear and well-organized research objective, and contributions.
2- The authors have compared their studies sample to a larger survey conducted in the same year by the BullDogJob.pl portal in Poland in order to verify the representativeness of the sample in relation to the target population, but there is no explanation of how they have done this. Please explain and add some justification regarding the variables and characteristics of the larger study and how your research has been adopted from it.
3- Tables 1-8 are too much. I recommend the authors try to merge some of them. In my opinion, it reads better if you do so but it is just a suggestion. Feel free to accept or reject.
4- The discussion section lacks a critical evaluation of the results and comparison with the existing studies. This section should be further expanded, highlighting opportunities and challenges and also future directions for further developments and studies.
5- The English of the manuscript should be checked for removing some minor typos and grammatical errors.
Author Response
Dear Reviewer,
Thank you very much for your rightful comments on our work, which we have taken into account as follows.
- Referring to your comments on the introduction, we have decided to rewrite this section, emphasising research objective and contribution of the research. In addition, we put the current study in the context of previous studies, which give us oportunity to better explain the paper aim.
- We have supplemented the study description with the characteristics of the survey to which we compared our research. The comparison itself made us aware that we were not in a position to conduct statistical analysis by individual characteristics because its lack of representativeness.
- In conclusion we presented evaluation of the results following the given hypothesis. This section was also expanded with opportunities and challenges and also future directions for further studies.
- The article has been verified by a proofreader, so misspellings and grammatical errors have been removed.
Thank you very much for your insightful review, which allowed us to make changes to improve the quality of our article. We are open to any further feedback.
Kind regards,
Milan Popović
Michał Błaszczyk
Karolina Zajdel
Radosław Zajdel

Round 2
Reviewer 1 Report
The paper has been modified. However the modifications are not sufficient given that the major revision was sought. In addition to the previous review comments that were not met i introduce the following points and improve the paper accordingly.
1. The paper introduces hypothesis in the section one, even though it should be presented after a comprehensive theoretical review , which is missing.
The literature is only reviewed in section 1.
Makokoane, J.K. (2022). The Dichotomy of a Changing Workplace: Analysing South Africa’s Newspapers during the Covid-19 Situation. International Journal of Management Science and Business Administration, 8(5), 26-40.
Su, R., Obrenovic, B., Du, J., Godinic, D., & Khudaykulov, A. (2022). COVID-19 Pandemic Implications for Corporate Sustainability and Society: A Literature Review. International journal of environmental research and public health, 19(3), 1592.
Onyeukwu, P.E., Madu, J.E. & Adeniyi, A. (2022). Teaching and Learning in Post-Covid-19 Era: An Evaluation of Digital Transformation Experience. International Journal of Management Science and Business Administration, 8(5), 41-56.
2. The summary tables of the qualitative study should be created. Pie charts are not neccessary, and can easily be summarized.
3. Results of the study should be elaborated in the context of newly added literature. Originality and contributions, innovative points of research should be enhanced.
Author Response
Dear Reviewer,
Thank you very much for your rightful comments on our work. Concerning the feedback we have received, we decided to significantly rewrite the first section, along with adding a section with a comprehensive theoretical review. With the consensus that pie charts take up too much space in relation to the cognitive value they offer, we decided to follow the advice and create a summary table. In the conclusions section, we have added an evaluation of the results with a comparison to the existing studies.
Thank you very much for your insightful reviews, which allowed us to make changes to improve the quality of our article. We are open to any further feedback.
Kind regards,
Milan Popović
Michał Błaszczyk
Karolina Zajdel
Radosław Zajdel

Reviewer 2 Report
The paper has apparently been revised by the authors, but I still do not feel it is of an appropriate standard for publication in a journal. Please do not include the name of the figures in the results section as a question in the questionnaire, but formulate it as a title. In any case, I would also ask for a nice citation of the source and the item number, as these are important for the interpretation of the results.
Author Response
Dear Reviewer,
Thank you very much for your rightful comments on our work. In order to improve our paper, we decided to significantly rewrite the first section, along with adding a section with a comprehensive theoretical review. We have observed that pie charts take up too much space in relation to the cognitive value they offer, so we decided to create a summary table instead. In the conclusions section, we have added an evaluation of the results with a comparison to the existing studies.
Thank you very much for your insightful reviews, which allowed us to make changes to improve the quality of our article.
Kind regards,
Milan Popović
Michał Błaszczyk
Karolina Zajdel
Radosław Zajdel

Reviewer 3 Report
The authors have not addressed my comments properly and I could not see any meaningful and considerable improvement. My comments have been ignored and touched upon very slightly. More in-depth consideration and efforts are needed from the authors' side.
Author Response
Dear Reviewer,
Thank you very much for your rightful comments on our work. Concerning both reviews we have received, we decided to significantly rewrite the first section, along with adding a section with a comprehensive theoretical review, as well as the conclusions section, where we have added an evaluation of the results with a comparison to the existing studies. We decided to leave the tables describing data similarities and believed that they should summarise each paragraph with the statistical verification of individual factors. However, we have observed that pie charts take up too much space in relation to the cognitive value they offer, so we decided to create a summary table instead.
Thank you very much for your insightful reviews, which allowed us to make changes to improve the quality of our article. We are open to any further feedback.
Kind regards,
Milan Popović
Michał Błaszczyk
Karolina Zajdel
Radosław Zajdel

Round 3
Reviewer 1 Report
The paper has been improved in the elements that were identified during the review. The paper could use a bit more polishing and proofreading, but overall improvements have been made.
Author Response
Dear Reviewer,
Thank you very much for your insightful reviews, which allowed us to make changes to improve the quality of our article.
In line with your comments, the article has been verified by a proofreader, so misspellings and grammatical errors have been removed
Kind regards,
Milan Popović
Michał Błaszczyk
Karolina Zajdel
Radosław Zajdel
Reviewer 3 Report
Thanks for revising the manuscript addressing my comments.
Author Response
Dear Reviewer,
Thank you very much for your insightful reviews, which allowed us to make changes to improve the quality of our article.
The article has been verified by a proofreader, so misspellings and grammatical errors have been removed.
Kind regards,
Milan Popović
Michał Błaszczyk
Karolina Zajdel
Radosław Zajdel